# Effects on Cell Growth, Lipid and Biochemical Composition of *Thalassiosira weissflogii* (Bacillariophyceae) Cultured under Two Nitrogen Sources

Francisco Eduardo Hernández-Sandoval [1,*], Jorge Arturo Del Ángel-Rodríguez [2,*], Erick Julian Núñez-Vázquez [1,*], Christine Johanna Band-Schmidt [3], Bertha Olivia Arredondo-Vega [1], Ángel Isidro Campa-Córdova [1], Manuel Moreno-Legorreta [1], Leyberth José Fernández-Herrera [3] and David Javier López-Cortés [1,†]

1    Centro de Investigaciones Biológicas del Noroeste (CIBNOR), La Paz 23096, Mexico; kitty04@cibnor.mx (B.O.A.-V.); angcamp04@cibnor.mx (Á.I.C.-C.); legoreta04@cibnor.mx (M.M.-L.)
2    Department of Ocean Sciences, Memorial University of Newfoundland, St. John's, NL A1K 3E6, Canada
3    Departamento de Plancton y Ecología Marina, Centro Interdisciplinario de Ciencias Marinas (IPN-CICIMAR), Instituto Politécnico Nacional, La Paz 23096, Mexico; cbands@ipn.mx (C.J.B.-S.); lfernandezh1200@alumno.ipn.mx (L.J.F.-H.)
*    Correspondence: fhernan04@cibnor.mx (F.E.H.-S.); jadar7@mun.ca (J.A.D.Á.-R.); enunez04@cibnor.mx (E.J.N.-V.)
†    Deceased.

**Abstract:** The protein and polyunsaturated fatty acid (PUFA) enrichment of microalgae can improve their nutritional value for larvae of various reared organisms. Protein enrichment can be achieved by increasing nitrogen concentration and selecting nitrogen sources that are easy to assimilate during microalga culture. Nitrogen-rich cultures can increase organism growth, biomass, and protein content, but their lipid content tends to stall. Since the diatom *Thalassiosira weissflogii* is usually provided to feed shrimp larvae, this study evaluated its digestibility and biochemical composition, culturing it with two nitrogen sources ($NaNO_3$ and $NH_4Cl$) at different concentrations (111.25, 222.50, 445 and 890 μM). The cell abundance, dry-weight biomass, Chl *a*, proteins, carbohydrates, total lipids and essential fatty acids were determined. The cell density and biomass peaked faster (day 12) with treatment < 890 μM than with 890 μM (day 15) in both nitrogen sources. However, the highest cell density, biomass and peak protein yield were not significantly different among treatments, suggesting the need to compare maintenance costs for a given production. After nine days of culture, concentrations ≤ 222.5 μM increased lipid content irrespective of the nitrogen source and decreased by 10–20% afterwards. With higher lipid production, the dominant PUFA were eicosapentaenoic acid (EPA) and docosahexaenoic acid (DHA). One gram of $NH_4Cl$ provides ~60% more nitrogen than 1 g of $NaNO_3$. In conclusion, based on time and growth rate, *T. weissflogii* cultivated with $NH_4Cl$ at 222.50 μM produced EPA and DHA at a better yield–cost ratio for biomass and lipid production. Furthermore, its nutritional value as enriched live-food for the reared larvae of marine organisms suggests potential biotechnological applications for aquaculture.

**Keywords:** biotechnological aquaculture; biochemical composition; fatty acid; nitrogen source; algal application

## 1. Introduction

In the last few years, interest in aquaculture has increased worldwide because of the growing market demand and high commercial value created by these products [1]. Their production success is undoubtedly related to the quality and quantity of the food supplied [2]. Microalgae are traditionally used in aquaculture for larval growth. Nevertheless, although attempts have been conducted to substitute natural food with microencapsulates, yeast or bacteria, the poor enzymatic activity and nonfunctional stomach of larvae do not

allow them to digest formulated diets completely [3,4]. Accordingly, natural microalgae are still considered as the main live food source for marine larvae [2,5].

As a consequence of their nutritional value, diatom species, such as *Skeletonema costatum*, *Thalassiosira* sp., *Phaeodactylum* sp., *Chaetoceros gracilis*, *C. calcitrans* and the *flagellates Pavlova lutheri*, *Isochrysis galbana*, *Tetraselmis chuii* and *T. suecica* are all grown as food for rearing marine organisms of commercial value [6,7]. Their biochemical composition is important for their nutritional value because of their ability to synthesize and accumulate essential fatty acids and other nutrients [8,9].

Among the species previously mentioned, *Thalassiosira weissflogii* has been provided successfully for feeding shrimp larvae [10] because of its shape, digestibility and biochemical composition [11], which can vary as a result of changes in the environment [12–14]. Thus, the effects of temperature, salinity, pH, nutrients and their interactions have been addressed [14–16]. Additionally, the growth phase is recognized as an important factor influencing the biochemical composition of microalgae. For example, during the stationary phase, when nitrogen becomes limiting, microalgae shift their metabolism to carbohydrate and lipid production [17]. Among all the physical-chemical factors, nitrogen influences lipid metabolism and its deficient conditions increase fatty acid concentration in various microalgae [18,19]. Nitrogen in the form of nitrate, ammonia or urea (or their combinations) is added to most media for commercial aquaculture in high concentrations and changes in nitrogen supply are known to strongly influence the growth and biochemical composition of microalgae [20]. Furthermore, nitrogen is easy to manipulate and less expensive compared to other growth factors. Comparatively few studies have analyzed the effects of different nitrogen sources on the biochemical composition of the diatom *T. weissflogii*. Therefore, this study evaluated different concentrations of two nitrogen sources, sodium nitrate ($NaNO_3$) and ammonium chloride ($NH_4Cl$), on the growth, biochemical composition (protein, carbohydrate and lipid) and main fatty acids in the microalga *T. weissflogii*.

## 2. Materials and Methods

### 2.1. Strain and Culture

*Thalassiosira weissflogii* (CIB-85 strain) was acquired by donation of CITMA (Ministerio de Ciencia y Tecnología y Medio Ambiente de Cuba; https://www.citma.gob.cu (accessed on 6 November 2021). The species was grown in batch cultures in 2.5 L Fernbach glass flasks (Thermo Fisher Scientific, Waltham, MA, USA) with f/2 + Si media [21]. Culture conditions were: illumination of 120 µmol photon $m^{-2}$ $s^{-1}$, continuous aeration, photoperiod of 12 h:12 h light/dark cycle, and temperature of $22 \pm 0.5\,°C$.

### 2.2. Experimental Design

Culture media f/2 + Si for experiments was initially prepared without nitrogen. The selected nitrogen source and concentration was added posteriorly (Table 1). Each experimental combination was assayed in quadruplicate cultures (i.e., $8 \times 4 = 32$). All treatments were inoculated with an initial concentration of $2.37 \times 10^5$ cells $mL^{-1}$. Cultures were progressively conditioned to each nitrogen concentration from an initial condition of 890 µM. The cells were adapted for three generations to each treatment before starting the batch culture. The experiment was conducted for 18 days.

### 2.3. Growth and Biomass Production

Cell density and dry biomass were monitored every third day using seven replicates for each sample (i.e., $8 \times 7 = 56$/day). Cell counts were performed on a Neubauer chamber (VWR SCIENTIFIC, superior, VWR International, Radnor, PA, USA).

Cell growth was evaluated using two models: the first one was via the logistic growth model, [22–24].

$$dN_t/dt = rN[(k - N_t)/k], \tag{1}$$

where $t$ is time (h); $N_t$ is the cell density (cells $mL^{-1}$) at time $t$; $r$ is the proportionality constant ($h^{-1}$); and $k$ is the carrying capacity of the system (cells $mL^{-1}$) or maximum cell

density ($N_{max}$) reached at the end of the exponential phase. The value $k/2$, which is half the carrying capacity (cells mL$^{-1}$), was also estimated, and $t_{k/2}$ represents the estimated time (h) to reach half of the carrying capacity. For this model, the non-linear regression parameters were estimated via the iterative process to reach the least sum of squares (SSQ).

**Table 1.** Sources and concentrations of nitrogen used in *Thalassiosira weissflogii* experimental culture media.

| Concentration (μM) | Treatments (NH$_4$Cl) | Treatments (NaNO$_3$) |
| --- | --- | --- |
| 111.25 | C1 | N1 |
| 222.5 | C2 | N2 |
| 445 | C3 | N3 |
| 890 | C4 | N4 |

Additionally, two other parameters were estimated. The first was maximum specific growth rate μ (h$^{-1}$), calculated by the equation [25,26]:

$$u_{max} = \ln(k/N_1)/(t_k - t_1), \tag{2}$$

where $t_k$ is time at the end of the exponential phase, and $t_1$ and $N_1$ are time and cell density at the beginning of the exponential phase, respectively. The second parameter was population doubling time (h), estimated as, [27]:

$$PDT = \ln 2/u, \tag{3}$$

The second model was an exponential rise to the maximum of the form: $N_t = N_0 + a\,(1 - e^{(-b \cdot t)})$, where cell density increase is represented by parameter $a$, and estimated growth rate is estimated by parameter $b$. For this model, time values were represented as days, thus at the beginning of the experiment:

$$(t = 0)\ N_t = N_0, \tag{4}$$

Nonlinear regressions were performed using the preset model for exponential rise to maximum, with three parameters, available in Sigmaplot Version 12.

Dry biomass was determined every 48 h by filtering 10 mL of culture through preweighed and pre-burned GF/F glass-fiber filter, 0.7 μm pore size (Whatman, Maidstone, UK). Biomass was further washed with 0.5 M ammonium formate to remove salt precipitates and dried at 60 °C for 24 h. The weight difference between the filter and the filter with the sample was determined.

### 2.4. Biochemical Composition

Biochemical composition was also monitored every 48 h, using seven replicates for each sample. A sample of 100 mL was obtained and centrifuged at 1658× *g* (BECKMAN GPR Centrifuge, CA, USA) for 15 min at 5 °C, and further lyophilized (LABCONCO 5.0 Liter, Virtis Freeze Dryers, Kansas City, MO, USA).

#### 2.4.1. Total Protein

Total protein content (5 mg) of lyophilized microalgae was determined spectrophotometrically with the Folin-Phenol Reagent (Sigma-Aldrich, St. Louis, MO, USA) [28]. Bovine serum albumin (BSA) was used as a standard (Sigma Aldrich STD). Samples were analyzed at 570 nm with a spectrophotometer (Beckman DU 640, spectrophotometer Beckman Coulter, Inc. Pasadena, CA, USA).

### 2.4.2. Total Carbohydrates

The carbohydrate content of 5 mg lyophilized microalgae was analyzed according to the sulfuric acid colorimetric method [29], based on phenolphthalein absorbance. Samples were analyzed at 490 nm (Beckman DU 640 spectrophotometer Beckman Coulter, Inc.).

### 2.4.3. Chlorophyll *a*

Two mg of lyophilized algae were extracted with 4 mL of acetone 100% HPLC grade (high performance liquid chromatography). Samples were stored at $-20$ °C every 24 h. Extract was recovered after centrifugation (1658× *g* at 5 °C for 15 min). Chlorophyll *a* (Chl *a*) was separated and quantified by HPLC (Mod. 1100, Hewlett Packard, CA, USA), as described in [30]. The mobile phase consisted of MeOH: 1 N aqueous ammonium acetate, 70:30% *v/v* (solvent A), and MeOH (solvent B). A C8 MOS Hypersil column 10 cm × 0.45 cm, 5 μm particle size was used with standards for identification and quantification. For the identification of pigments, retention time standards (RST) (International Agency for $^{14}$C determinations, Denmark, DHI-Denmark, PPS-CHLA) and absorption spectra (350–750 nm) were considered.

### 2.4.4. Total Lipids

The total content of lipids were extracted according to the method [31], where 5 mg of lyophilized microalgae was added to 2 mL chloroform: methanol (2:1 *v/v*) HPLC grade at 4 °C for 24 h. Extracts were centrifuged at 1658× *g* for 20 min, and the supernatant was collected. Distilled water was added until the system became biphasic. The lower chloroform layer was separated and dried under nitrogen ($N_2$) flow. Lipids were quantified accordingly [32], which involved organic matter carbonization with sulfuric acid, using triestearine as the standard. Samples were analyzed at 375 nm with a spectrophotometer (Beckman DU 640, Beckman Coulter, Inc.).

### 2.4.5. Fatty Acid Composition and Statistical Analyses

Methanolysis of fatty acids with HCl:$CH_3OH$ (5:95 *v/v*) was performed as described in [33]. Methyl ester extract was evaporated to dryness under a stream of $N_2$. The resulting methyl esters were resuspended in *n*-hexane HPLC. The extract was injected into a gas chromatograph mass spectrometer with an electronic ionization detector (GCD Plus Hewlett Packard, model G1800B) equipped with an Omega Wax Column (30 m × 0.25 mm × 0.25 μm). Oven temperature was programmed as follows: initially 110 °C with 280 °C as the maximum temperature. The flame detector temperature was 260 °C. The carrier gas utilized was helium at a flow rate of 0.9 mL min$^{-1}$. The identification of fatty acids was performed using two criteria: (1) retention time of methylated standard and (2) presence of the characteristic ion mass/charge (*m/z*). Different fatty acid methyl esters (FAME chemical standard, Sigma Aldrich) were used to determine retention time and response factor of each fatty acid.

The results for biochemical composition and cell growth were analyzed by a multifactorial analysis of variance (ANOVA) considering nitrogen source, nitrogen concentration and time as factors. The level of significance was $\alpha = 0.05$ [34]. All analyses were carried out using STATISTICA® software v.6.0 (Statsoft, Tulsa, OK, USA).

## 3. Results

### 3.1. Growth and Biomass Production of Microalgae

As expected, the *T. weissflogii* peak cell density (*k*) improved significantly ($p < 0.05$) when the nitrogen concentration increased from 111.25 to 890 μM. According to the sigmoidal curve model (Table 2), the highest $NaNO_3$ (N4) concentration produced the largest cell density (>12.5 × 10$^6$ cells mL$^{-1}$) at the end of the exponential phase. However, the highest $NH_4Cl$ (C4) concentration yielded a similar cell density (~6% less) but with a higher proportionally constant (*r*) (0.385 vs. 0.275 h$^{-1}$). Accordingly, the estimated time to reach half of the maximum cell density ($t_{k/2}$) occurred 33 h earlier for C4 compared with N4. In general, *T. weissflogii* reached $t_{k/2}$ 7–30 h earlier when the nitrogen source was $NH_4Cl$.

The exponential rise model (Table 3, bottom) also confirmed higher final cell densities (*a* parameter) for N4 and C4, but they were not significantly different. Within this model, the estimated growth rate (*b* parameter) was not significantly different among similar concentrations of different nitrogen sources.

**Table 2.** Parameter estimations of *Thalassiosira weissflogii* growth curves with two different nitrogen sources at different concentrations via different models.

| | NaNO₃ μmol | | | | NH₄Cl μmol | | | |
|---|---|---|---|---|---|---|---|---|
| Treatment | N1 | N2 | N3 | N4 | C1 | C2 | C3 | C4 |
| μmol L$^{-1}$ [as Salt] | 111.25 | 222.50 | 445.00 | 890.00 | 111.25 | 222.50 | 445.00 | 890.00 |
| μg L$^{-1}$ [N] | 18.33 | 36.67 | 73.33 | 146.67 | 29.13 | 58.26 | 116.52 | 233.05 |
| | Parameter estimation of the sigmoidal curve via least squares (SSQ) | | | | | | | |
| $N_0$ (cells mL$^{-1}$) | 507,498 | 416,176 | 505,885 | 457,116 | 473,424 | 439,717 | 485,535 | 426,034 |
| $r$ (h$^{-1}$) | 0.0304 | 0.0221 | 0.0351 | 0.0275 | 0.0358 | 0.0294 | 0.0367 | 0.0385 |
| $k$ or $N_{max}$ (cells mL$^{-1}$) | 5,378,478 | 8,360,922 | 9,553,601 | 12,580,306 | 5,589,747 | 7,700,407 | 7,921,201 | 11,852,018 |
| % max $k$ | 43 | 66 | 76 | 100 | 44 | 61 | 63 | 94 |
| $k/2$ (cells mL$^{-1}$) | 2,689,239 | 4,180,461 | 4,776,801 | 6,290,153 | 2,794,873 | 3,850,203 | 3,960,600 | 5,926,009 |
| $t_{k/2}$ (hours) | 74 | 134 | 82 | 119 | 67 | 95 | 75 | 86 |
| | Formula estimation of $\mu$ and *PDT* | | | | | | | |
| $\mu_{max}$ (h$^{-1}$) | 0.0111 | 0.0125 | 0.0133 | 0.0114 | 0.0117 | 0.0130 | 0.0129 | 0.0118 |
| *PDT* (hours) | 62.2 | 55.3 | 52.2 | 60.8 | 59.4 | 53.3 | 53.9 | 58.8 |
| | Parameter estimation of an exponential rise to maximum model via non-linear regression (Note *t* (days) | | | | | | | |
| $b$ | 0.343 | 0.114 | 0.241 | 0.122 | 0.277 | 0.173 | 0.295 | 0.145 |
| 95% CI $b$ | 0.264 | 0.064 | 0.087 | 0.040 | 0.173 | 0.080 | 0.135 | 0.044 |
| $a$ | 4,349,873 | 9,604,878 | 9,306,673 | 15,151,914 | 5,246,430 | 8,220,818 | 7,207,902 | 14,152,484 |
| 95% CI $a$ | 851,406 | 2,393,787 | 1,012,272 | 2,163,974 | 931,908 | 1,363,124 | 902,546 | 1,691,236 |

Values represent equations 1 and 2 parameters and therefore describe the whole growth curves.

**Table 3.** Total protein, carbohydrate and lipid contents (mean value and standard deviation) of *Thalassiosira weissflogii* dry biomass at different days, cultured at four different nitrogen concentrations of NaNO₃.

| | | Time (Days) | | | |
|---|---|---|---|---|---|
| | NaNO₃ μmol | 3 | 9 | 15 | 18 |
| Protein (μg mg$^{-1}$) | 111.25 | 74.7 ± 9.5 | 88.5 ± 7.1 | 77.5 ± 12.7 | 45.2 ± 7.7 |
| | 222.50 | 158.9 ± 12.0 | 123.7 ± 43.4 | 116.6 ± 19.1 | 96.27 ± 2.5 |
| | 445 | 325.1 ± 63.4 | 200.0 ± 24.4 | 183.4 ± 2.2 | 144.7 ± 7.5 |
| | 890 | 351.4 ± 22.8 | 239.7 ± 36.5 | 232.9 ± 3.6 | 185.6 ± 10.5 |
| Carbohydrates (μg mg$^{-1}$) | 111.25 | 178.0 ± 27.7 | 300.4 ± 9.7 | 399.4 ± 50.3 | 290.2 ± 9.9 |
| | 222.50 | 188.8 ± 70.5 | 386.7 ± 10.9 | 312.8 ± 48.7 | 436.6 ± 5.3 |
| | 445 | 216.5 ± 66.7 | 274.1 ± 36.7 | 334.1 ± 43.0 | 272.1 ± 64.7 |
| | 890 | 215.8 ± 15.4 | 253.7 ± 10.0 | 238.5 ± 7.4 | 211.5 ± 5.8 |
| Lipids (μg mg$^{-1}$) | 111.25 | 318.6 ± 22.0 | 494.5 ± 10.5 | 447.1 ± 20.9 | 370.1 ± 8.7 |
| | 222.50 | 338.9 ± 31.2 | 460.4 ± 51.1 | 388.2 ± 22.4 | 356.4 ± 40.2 |
| | 445 | 244.8 ± 5.7 | 349.1 ± 44.2 | 340.3 ± 38.7 | 268.5 ± 32.8 |
| | 890 | 240.0 ± 10.1 | 281.9 ± 61.9 | 335.5 ± 12.5 | 289.1 ± 13.3 |

Diatom *T. weissflogii* treated with NaNO₃ and NH₄Cl concentrations of 222.5 and 445 μmol L$^{-1}$ (N2, N3, C2 and C3), respectively, achieved population duplication times (*PDT*) < 56 h, which suggests that moderate salt concentrations provided concentrations close to the optimum nitrogen amounts required for cell duplication, i.e., neither limited nor in excess. With the 890 μM treatment, NaNO₃ reached a plateau in biomass production (~0.4 mg mL$^{-1}$) around day 12 and remained at similar levels until day 18. By contrast, the treatment with 890 μM NH₄Cl reached a similar peak value (~0.4 mg mL$^{-1}$) until day 15 but rapidly decreased at day 18. The rest of the treatments did not exceed a biomass concentration of 0.250 mg mL$^{-1}$.

The cell density (cell mL$^{-1}$) correlated positively with the dry weight measurements (mg mL$^{-1}$) for both nitrogen sources. For NaNO₃, the slope was $2.45 \times 10^{-8}$ mg cell$^{-1}$,

$R^2 = 0.93$ and $p < 0.05$, whereas for $NH_4Cl$ the slope was $2.53 \times 10^{-8}$ mg cell$^{-1}$, $R^2 = 0.95$ and $p < 0.05$. Furthermore, no significant differences were observed between slopes for both sources (ANCOVA Source $p = 0.172$).

### 3.2. Biochemical Composition

#### 3.2.1. Chlorophyll *a*

The chlorophyll *a* content in *T. weissflogii* increased with time and nitrogen concentration ($p < 0.001$). However, the time to reach peak production varied depending on the source and concentration. For instance, $NaNO_3$ and $NH_4Cl$ treatments with 111.25 μM reached small peak chlorophyll *a* (2500–3900 ng mg$^{-1}$ dry weigh) values earlier (day 12). By contrast, treatments with 890 μM yielded the highest chlorophyll *a* production, with similar results at peak production (7084–7199 and 7039–8691 ng mg$^{-1}$ dry weight for $NaNO_3$ and $NH_4Cl$, respectively). However, with $NaNO_3$, the peak was reached at the end of the experiment (day 18), whereas for $NH_4Cl$, it was reached three days earlier (day 15).

#### 3.2.2. Proteins, Carbohydrates and Lipids

As expected, with both salts, a clear tendency was observed towards an increase in protein content as a consequence of an increase in nitrogen concentration ($p < 0.05$). However, this difference was not sustained throughout the experiment. This result is common in batch cultures without nitrogen replenishment. As the microalgae grew, moderate-to-strong nitrogen depletion decreased amino acid synthesis, as was expected. For instance, at the onset of the experiment (day 3), a significantly high protein concentration (~390 μg mg$^{-1}$) was observed in the 890 μM-treated $NH_4Cl$ and $NaNO_3$, as well as in the 445 μM-treated $NaNO_3$. As the experiment progressed, protein concentration decreased, reaching concentrations of <220 μg mg$^{-1}$ by day 18. A high decrease, from 47 to 57%, in protein content was observed in very low (N1 and C1), moderately high (N3 and C3) and high (N4 and C4) nitrogen salt concentrations with time. Moderately low salt concentrations (C2 and N2) initially yielded lower protein concentrations but showed less variation with time. In this case, the decline in protein concentration during the experiment was 39% for $NaNO_3$ and 31% for $NH_4Cl$ (Tables 3 and 4).

**Table 4.** Total protein, carbohydrate and lipid contents (mean value and standard deviation) of *Thalassiosira weissflogii* dry biomass at different days cultured at four different nitrogen concentrations of $NH_4Cl$.

| | | Time (Days) | | | |
|---|---|---|---|---|---|
| | NH$_4$Cl μmol | 3 | 9 | 15 | 18 |
| Protein (μg mg$^{-1}$) | 111.25 | 126.5 ± 10.5 | 72.21 ± 20.9 | 124.4 ± 16.6 | 102.9 ± 11.8 |
| | 222.50 | 143.4 ± 22.8 | 146.8 ± 11.8 | 165.8 ± 5.8 | 128.5 ± 23.3 |
| | 445 | 230.2 ± 16.0 | 238.3 ± 52.1 | 171.2 ± 43.7 | 183.8 ± 56.7 |
| | 890 | 393.1 ± 68.5 | 242.3 ± 12.5 | 218.7 ± 6.9 | 199.4 ± 11.5 |
| Carbohydrates (μg mg$^{-1}$) | 111.25 | 350.9 ± 45.9 | 354.3 ± 22.2 | 279.9 ± 45.6 | 279.1 ± 22.9 |
| | 222.50 | 351.7 ± 106.9 | 385.6 ± 13.6 | 380.9 ± 29.1 | 266.6 ± 21.4 |
| | 445 | 205.1 ± 77.4 | 348.6 ± 20.3 | 284.2 ± 20.6 | 248.1 ± 16.9 |
| | 890 | 142.3 ± 50.6 | 353.0 ± 54.3 | 271.6 ± 34.8 | 267.9 ± 30.0 |
| Lipids (μg mg$^{-1}$) | 111.25 | 378.6 ± 10.5 | 497.7 ± 11.1 | 404.7 ± 17.7 | 387.6 ± 11.1 |
| | 222.50 | 48.6 ± 28.3 | 450.9 ± 28.2 | 404.8 ± 0.08 | 425 ± 14.4 |
| | 445 | 263.1 ± 10.8 | 328.6 ± 24.3 | 414.2 ± 12.0 | 309.4 ± 7.2 |
| | 890 | 235.0 ± 9.4 | 319.8 ± 6.54 | 318.8 ± 9.0 | 309.2 ± 14.9 |

The carbohydrate and lipid contents were influenced by the interaction between nitrogen concentration and culture age ($p < 0.005$) and not by the nitrogen source. The carbohydrate concentration was the least consistent in time and nitrogen concentration. For instance, during most of the experiment, carbohydrates showed higher concentrations with high $NH_4Cl$ concentrations (C3 and C4) but no clear pattern was observed with time.

By contrast, the NaNO$_3$ cultures increased their carbohydrate concentration with time and increased, on most days, with low nitrogen concentration (Table 3).

As expected, the lipids reached highest concentrations at lower nitrogen concentrations. In the case of NaNO$_3$, they reached a plateau of ~460 µg mg$^{-1}$ for N2 and 495 µg mg$^{-1}$ for N1 after day 6. These values were not significantly different from those observed at day 9 or 12, suggesting a relatively long stationary phase in terms of lipid yield. For the treatments with NH$_4$Cl, C1 and C2 reached a small plateau of ~500 and 450 µg mg$^{-1}$ after day 9 and remained at similar values after day 12. In most treatments, the lipid content dropped after day 15, indicating a time limit for harvesting if maximum lipid concentrations are the main goal.

### 3.3. Fatty Acids

The *T. weissflogii* fatty acid pattern was divided based on the degree of saturation, namely SAFA (saturated fatty acid), MUFA (monounsaturated fatty acid) and PUFA (polyunsaturated fatty acid). During exponential growth, the microalgae tended to show similar fatty acid percentages between the two different nitrogen sources, but the fatty acid concentration differed ($p > 0.05$).

In the NaNO$_3$ cultures, SAFA constituted 53.52%, PUFA 44.78% and, in a lower proportion, MUFA contributed 1.69%. In the NH$_4$Cl cultures, SAFA was 50.16%, PUFA 48.60% and, in a lower proportion, MUFA contributed 1.22%. The *T. weissflogii* concentration of selected PUFAs varied widely as a consequence of nitrogen source and concentration (Table 5).

**Table 5.** Polyunsaturated fatty acid (PUFA), eicosapentaenoic acid (EPA) and docosahexaenoic acid (DHA) composition of *Thalassiosira weissflogii* growth in different nitrogen sources (µM) and concentrations in pg cells$^{-1}$ (mean ± SD). SD: standard deviation.

| | NaNO$_3$ (µM) | | | | NH$_4$Cl (µM) | | | |
|---|---|---|---|---|---|---|---|---|
| PUFA | 111.25 | 222.5 | 445 | 890 | 111.25 | 222.5 | 445 | 890 |
| 18:2 | 1.20 ± 0.53 | 1.59 ± 0.46 | 1.32 ± 0.31 | 2.96 ± 0.83 | 0.63 ± 0.21 | 0.97 ± 0.38 | 1.38 ± 0.68 | 1.52 ± 0.37 |
| 18:3 | 3.03 ± 1.13 | 2.82 ± 1.17 | 2.58 ± 1.91 | 4.08 ± 2.69 | 1.33 ± 0.29 | 2.57 ± 1.99 | 3.53 ± 1.29 | 4.08 ± 2.69 |
| 18:4 | 3.11 ± 1.05 | 4.86 ± 1.82 | 3.53 ± 2.13 | 5.95 ± 4.67 | 3.61 ± 1.43 | 3.14 ± 1.90 | 5.83 ± 3.61 | 1.89 ± 1.45 |
| 20:4 | 0.014 ± 0.04 | 0.026 ± 0.01 | 0.059 ± 0.04 | 0.043 ± 0.02 | 0.01 ± 0.0009 | 0.01 ± 0.006 | 0.01 ± 0.007 | 0.02 ± 0.009 |
| 20:5n3 (EPA) | 16.67 ± 4.35 | 7.58 ± 2.48 | 14.12 ± 3.09 | 7.29 ± 1.14 | 11.94 ± 9.47 | 16.91 ± 4.73 | 10.24 ± 3.36 | 11.04 ± 6.67 |
| 22:6n3 (DHA) | 7.22 ± 1.45 | 4.37 ± 3.02 | 6.45 ± 2.76 | 9.24 ± 3.69 | 4.82 ± 2.06 | 8.38 ± 2.01 | 5.25 ± 2.44 | 6.77 ± 4.16 |

The major fatty acids were eicosapentaenoic (EPA, 20:5n3) and docosahexaenoic (DHA, 22:6n3) acids, and both increased at higher NaNO$_3$. EPA concentration was higher than DHA in treatments with low and medium NaNO$_3$ concentrations. However, at the highest NaNO$_3$ concentration, the DHA concentration surpassed that of EPA. By contrast, the NH$_4$Cl treatments yielded mixed results: the EPA and DHA concentrations were higher at moderately low and very high NH$_4$Cl concentrations, whereas at very low and moderately high NH$_4$Cl concentrations, these values dropped. In all four NH$_4$Cl concentrations, EPA concentration surpassed that of DHA.

## 4. Discussion

### 4.1. Microalgae Growth and Biomass Production

As the nitrogen concentration increased in the cultures, the *T. weissflogii* biomass concentration and cell density increased. A decrease in algal biomass concentration in low nitrate concentration was also observed [35] in *Nannochloropsis* sp.

As previously described, reduced nitrogen forms as ammonium or urea are preferably used for microalgae, since nitrate must be transformed into ammonium before it is utilized assimilated by algal cells, so more energy expenditure is necessary [36,37]. This assertion remains true for lower nitrogen salt concentrations and was particularly evident for moderately low concentrations (N2 and C2), where the NH$_4$Cl yielded a biomass peak twice as large as that yielded by the NaNO$_3$. However, high NH$_4$Cl concentration cultures reached

peak biomass three days later than similar NaNO$_3$ cultures, which suggests that at high salt concentrations, NaNO$_3$ can be assimilated slightly faster than NH$_4$Cl.

Considering only the cost per metric ton of each salt (\$200–\$430 USD ton$^{-1}$ NaNO$_3$ vs. \$90–\$200 USD metric ton$^{-1}$ NH$_4$Cl) [38,39] and the amount required to run the experiment, an estimated production (mg USD$^{-1}$) was computed. Since NH$_4$Cl is less than half the price of NaNO$_3$, treatments with low NH$_4$Cl were the most cost-effective in all cases. All the treatments reached their most cost-effective values after 15 or 18 days. However, differences in time and growth rate were also important for estimated costs. For instance, C1 and C2 (low and moderately low NH$_4$Cl) provided a maximum of >6 mg DW USD$^{-1}$ after 15 days, whereas C3 and C4 (moderately high and high NH$_4$Cl) provided a maximum of 3–4 mg DW USD$^{-1}$ after 15 and 18 days, respectively. By contrast, N1 reached a maximum of 3.2 mg DW USD$^{-1}$ after 18 days, and N2–N4 provided only 1.44–1.68 mg DW USD$^{-1}$ after 15 or 18 days.

*4.2. Biochemical Composition*

Since microalgal species can vary in nutritional value as a function of culture conditions, several studies have attempted to manipulate them with the objective of increasing biochemical products [2,40,41], protein [13], carbohydrate [35,42], lipid [43–45] and fatty acid [16,46,47] contents. Many microalgal species, respond to nitrogen deficiency by increasing their lipid content. A lack of nitrogen and carbon availability allows microalgae to switch from synthetizing proteins to lipids, with correspondingly slower growth rates and lower biomass production [48]. Some strains of the genus *Thalassiosira* have been described as potentially harmful [49] but the production of allelochemical compounds from *T. weisflogii* strains with allelopathic effects against potentially harmful microalgae, such as *Phaeocystis globosa*, has also been reported [50].

Chlorophyll is utilized as an intracellular nitrogen pool to support further cell growth and biomass production as nitrogen in the media becomes depleted [47,51]. Chlorophyll *a* follows a similar trend to biomass production, since chlorophyll *a* is a nitrogen-rich compound that contains four nitrogen atoms. This suggests that cells accumulate large quantities of chlorophyll molecules when a nitrogen source is abundantly available [52]. This pattern has been described in a previous research [13].

The protein content revealed maximum values in the exponential growth phase, decreasing throughout the stationary phase. Costard [13] found the same trend for five genera of microalgae from the Brazilian coast (*Bellerochea* sp., Van Heurck 1885; *Chaetoceros* sp., Ehrenberg 1844; *Chlorella* sp. M. Beijerinck 1890; *Rhodomonas* sp. Karsten 1898 and *Thalassiosira* sp. Cleve 1873). Silva [53] also reported a peak in the protein content of *Rhodomonas* in the transitional growth phase and a declination in the stationary stage.

Generally, as microalgae enter the stationary stage due to limited nutrients, cells accumulate carbon in the form of lipids and carbohydrates [13,54]. Navarro [55] observed that the lipid content tends to increase when cultures reach the stationary phase, especially when silicate and nitrogen are limited. This study proved that lipid content rises as nitrogen concentration declines in the cultures around day nine, i.e., at the onset of the stationary phase. This phenomenon was clearly observed in cultures with lower nitrogen concentrations (111.25, 222.50 μM). Lv [56] studied the effect of KNO$_3$ concentration on the lipid concentration. Different KNO$_3$ concentrations of 0.2, 1.0, 3.0 and 5.0 mM were applied to *Chlorella vulgaris* cultures and demonstrated that the lipid content decreased as the KNO$_3$ concentration increased.

*4.3. Fatty Acids*

The use of microalgae as natural sources of fatty acids for aquaculture has become the focus of scientific developments in recent years, particularly PUFA content, which performs specific physiological functions as a phospholipid biomembrane component [8].

In this study, the fatty acid content of *T. weissflogii* fatty was predominantly represented by SAFA and PUFA, which increased with lower nitrogen concentration cultures and culture



age. The fatty acid component can be modified by the nitrogen concentration and source. The presence of $NH_4^+$ stimulates the activity of desaturases, which catalyzes the formation of double bonds, forming unsaturated fatty acids; the highest PUFA levels were noted with the $NH_4Cl$ source. The major fatty acids were eicosapentaenoic acid (EPA, 20:5n3) and docosahexaenoic (DHA, 22:6n3). EPA results have been reported for *T. weissflogii* and other diatom strains as one of the major fatty acid contents [57–60].

## 5. Conclusions

Higher nitrogen concentrations influenced *T. weissflogii* cell metabolism, increasing cell biomass and protein content. By contrast, low nitrogen concentrations increased lipid concentration, probably due to their storage of cell reserves. In most cases, the carbohydrate concentrations also increased with nitrogen limitation.

With respect to production, moderately low $NH_4Cl$ treatment (C2) proved to be highly efficient in EPA, moderately efficient in DHA, and the most cost-effective in terms of biomass; no variations were observed in protein content with culture age. Concentrations of $NH_4Cl$ close to 222 $\mu M$ were considered the best option for cultivating *T. weissflogii* as food for the growth of marine larvae of commercial interest. However, if the goal is to maximize DHA production, $NaNO_3$ at 890 $\mu M$ should be considered for rearing purposes.

**Author Contributions:** Conceptualization, methodology, investigation and writing—original draft preparation, F.E.H.-S.; formal analysis and writing—review and editing J.A.D.Á.-R.; supervision, writing—review and editing E.J.N.-V.; writing—review and editing, C.J.B.-S.; resources and writing—review and editing B.O.A.-V.; writing—review and editing Á.I.C.-C.; data curation and software, M.M.-L.; visualization and editing L.J.F.-H.; conceptualization, writing—original draft preparation and funding acquisition, D.J.L.-C. All authors have read and agreed to the published version of the manuscript.

**Funding:** This research was funded by the institutional projects PC 0.11, PC 0.12 and AC0.8 (CIBNOR) and the student fellowship CONACyT 126348.

**Institutional Review Board Statement:** Not applicable for studies not involving humans or animals.

**Informed Consent Statement:** Not applicable.

**Data Availability Statement:** Not applicable.

**Acknowledgments:** The authors thank Laura Carreon-Palau for her assistance in the GC-MS analysis; Gerardo García for his help in the preparation of the figures; W. Johnson and Miguel Cordoba-Matson for improving the English version; and Diana Fischer for English editing.

**Conflicts of Interest:** The authors declare no conflict of interest.

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
