# Peer review of "Effects on Cell Growth, Lipid and Biochemical Composition of Thalassiosira weissflogii (Bacillariophyceae) Cultured under Two Nitrogen Sources"

_applsci, doi:10.3390/app12030961_

Round 1

Reviewer 1 Report

# Review Sandoval 2021   The project investigated the effects of two nitrogen sources on the cell growth, and biochemical composition (with a particular interes in lipids and fatty acids) of a diatom species. Comparing the effects of different nitrogen sources at different concentrations allowed the authors to comment on optimized growth conditions for omega-3 fatty acid production.     While the project topic has a lot of applied value, the manuscript in its current form does not do the research justice and could be improved in several ways.   ## General comments   The main strength of this manuscript is the comparisons made between diatom biochemical composition with changes in  nitrogen sources and concentrations.     However, for any comparison and related interpretations to be useful, it should be very clear to the reader on what basis these comparisons were made.  In this study the cultures were grown in batch (over a number of days) with different nitrogen sources and concentrations.  Questions that arise include:   1. do the cell numbers correlate with dry weight measurements (e.g. for the different nitrogen sources tested); if so, these could be fruitfully interchanged but if not, some of the current conclusions should be revisited and it should be considered whether there is a preference between presenting data per cell number or per dry weight (currently both are used)  2. are average or peak numbers used? 3. if so, why were they chosen 4. if so, please state clearly at each comparison made and conclusion drawn   Please state clearly what considerations were taken into account when recommending the "best option" for a nutrient source that would result in  a certain omega-3 fatty acid composition (as mentioned in the Abstract and Conclusion) e.g. was time and therefore growth rate and production rate taken into account or not?     ## Specific comments   line 17 & 22: please write out any abbreviation at first mention throughout the document   line 18: what is meant by "larvae's organisms"?   line 18 - 19: Please rewrite the following sentence in a clearer way:  "Nitrogen-rich cultures can increase its growth, biomass, protein content, but their lipid content tends to stall."  Without background knowledge the reader cannot interpret and correlate what "nitrogen-rich" has to do with increase in the parameters mentioned; or what it means that "lipid content stalls"   line 23 - 24:  "Treatments < 890 μM" implies ALL conditions tested for this study so how can a reader interpret the statement of:  "resulted in higher cell density and biomass"   line 25: missing word?   line 26: what does "afterwards at day 12" mean?   line 26: typo "Whit"   line 26 and 27:  Regarding EPA and DHA, see comment above about abbreviations   line 27:  On what basis is the statement made that sodium nitrate has more moles of nitrogen than ammonium chloride?  One mole sodium nitrate salt will provide one mole of nitrogen and the same for ammonium chloride salt.   line 44-45: genus and species names in italics   line 44: missing word?  "As consequence of their nutritional value"    line 46: genus and species names in italics   line 47: from the text is not clear what is meant by "valuable marine organisms"   line 56: in mentioning the importance of the growth phase influencing the biochemical composition, please elaborate for the sake of the reader for example by providing an example   line 59:  missing word?  "in various micro-algae increase their concentration in nitrogen deficient conditions"   line 65: typo,  diatom   line 74: looking up the f/2 media online it appears to already contain sodium nitrate.  please elaborate or explain how this was taken into account during the present study   line 75 - 76:  this sentence needs a verb   line 79:  perhaps consider "inoculated" instead of "initiated"   line 80: typo—maintained   line 81:  the section subheading mentions growth and biomass while information about biochemical composition is interspersed through the following paragraphs (e.g. lines 89 - 90). Please help the reader by either adjusting the subheading or creating a new subheading to mention the specifics regarding biochemical sample preparation   line 82: typo. "third"   line 84 - 85:  what is meant by "biochemical concentration"?   line 85:  line 82 mentions every third day while line 85 mentions every 48 hours. How should the reader think about this?   line 87: additional word? "washed with 0.5 M with ammonium"   line 88: perhaps consider replacing the word "duration" with "for"   lines 94 - 96 and equation one:  the units provided for the different variables in equation one does not appear to balance out.    line 94: typo;  cell density   line 96:  how are the concepts of maximum cell density compatible with occurring during the exponential phase?   line 103:  something missing between brackets?   line 103: typo;  "parameter"   line 102: it is not clear why an abbreviation for protein is provided    line 112: please provide some indication in the text that the Lowry was used and whether a kit was used or whether the reagents were prepared in-house   line 113: word missing?  "spectrophotometrically analysis"   line 123:  see comment above about alternative suggested for use of the word "during"   line 137:  please clarify what is meant by "lipids were determined".  does it refer to quantification, characterization, etc?   line 152:  typo; "FAME" (it is an abbreviation)   lines 161-162: a comparison is made between different nitrogen concentrations but please specify what other conditions were considered e.g. was it at the end of the experiment, at the peak, was it the average etc.   line 172:  Table 1 describes "Parameter estimations of growth curves".  Similar to the comment above it will help the reader to clarify which values were considered e.g. stationary phase? peak? average? minimum (e.g. in the case of PDT?)   lines 161 - 171:  while the data is currently presented as a table, relevant data could perhaps be shown as a figure? In this way the reader will easily be able to see relevant stages at a glance e.g. exponential phase, stationary phase, increases and declines, growth rates, lag phases etc   header of Table 1:  units of nitrogen sources given as micromol; is this correct?   Table 1:  how can the moles of nitrogen be different from the moles of salt for sodium nitrate and ammonium chloride?   table 2 caption: typo; "deviation"   lines 188 - 189:  please help the reader to understand and interpret the significance of this sentence   line 190 - 191:  it is hard to interpret "showed maximum values at 190 day 12> 2500 and < 3900 Chla". Perhaps some brackets or words or punctuation can help?   line 199: can the authors comment on the substantial drop in protein concentration after day 3 for sodium nitrate?   table 3 caption: typo; "deviation"   line 236:  what does the "Mean" refer to in the caption of Table 4?  the mean over the full duration of the experiment?  Please clarify.   line 238- 240:  please connect the abbreviations of EPA and DHA to nomenclature used in Table 4 right here in the text e.g. move the sentence from section 4.3    line 247:  what is meant by "growth". Growth rate? biomass?   lines 247 - 252:  three sentences almost saying the same thing three times.  Please remove the redundancy to focus on the most salient take-home message.     line 263-264: please provide a source and related point in time for which the prices are valid   lines 267 - 269:  please provide some justification why day 15 was chosen for comparison   line 277:  please explain what is meant by "lipid productivity"   line 306 - 307:  this statement contradicts the values provided in the Results section   line 324: please verify the unit provided    

Reviewer 2 Report

Technical mistakes:

Lines 44-46: all Latin names of genera and species have to be in Italic: e.g. Thalassiosira sp., but Skeletonema costatum

Line 48:  the biochemical composition plays an important role in the nutritional value - perhaps,would be better to change to: biochemical composition is important for the nutritional value

Line 53: "However, factors such as;" - Please, take away ";" 

Line 161: "As expected,. cell density of" - Please, change to "As expected,"

Line 312: "The results were similar to those found in other Pavlova species" - why "other"?, the authors do not discuss Pavlova genus 

Comments on the text:

Since the alga is used as a food, please, provide a discussion on the toxicity/no-toxicity of the species and its relations with red tides (For example, Thalassiosira weissflogii is non-toxic but is often associated with assemblages that form blooms in red tides (Bricelj et al. 1991, Yamaoka et al. 1998).

The citation of Marella TK, Tiwari A. Marine diatom Thalassiosira weissflogii based biorefinery for co-production of eicosapentaenoic acid and fucoxanthin. Bioresour Technol. 2020 Jul;307:123245. doi: 10.1016/j.biortech.2020.123245.  is missing

Line 44-46: The spesies "Skeletonema costatum, Thalassiosira sp., Phaeodactylum sp., Chaetoceros gracilis, C. calcitrans" are diatoms, and this has to be mentioned when they are enlisted with other algae, such as "the flagellates Pavlova lutheri, Isochrysis galbana, Tetraselmis chuii, and T. suecica", which are also microalgae but represent different from diatoms (Bacillariophyceae of Ochrophyta) algal lineages (e.g. Haptophyta, Chlorophyta..). And since in the whole text there is the same mixture of different algae, I would recommend to the authors to make comparisons with cultures of other diatoms, or of the same diatom (e.g. Marella et al. 2020) and not with microalgae from other taxonomic groups like Nannochloropsis, Rhodomonas, Pavlova, Chlorella, etc. Or, if such comparisons have to be done, they should be only additional since there is no doubt that the taxonomic position is important for the biochemical compounds, and then comes the influence of culture conditions.

The paper provides valuable information and, therefore such changes and editing are proposed to the authors. 

Reviewer 3 Report

Dear authors,

The major issue that does not work is presentation of results, particularly, Tables 2 and 3 (should include percentiles) needs to be consolidated and in the same section 3.2., Biochemical composition. It is important that Materials and Results flow systematically. Furthermore there are some issues with English language (grammar and spelling) that could be easily corrected if you can find relevant source to proof the text.

Good luck!

Round 2

Reviewer 1 Report

The authors addressed my concerns satisfactorily.